# Therapeutic efficacy of artemether-lumefantrine, artesunate-amodiaquine and dihydroartemisinin-piperaquine in the treatment of uncomplicated *Plasmodium falciparum* malaria in Sub-Saharan Africa: A systematic review and meta-analysis

Karol Marwa[1]*, Anthony Kapesa[2], Vito Baraka[3], Evelyne Konje[4], Benson Kidenya[5], Jackson Mukonzo[6], Erasmus Kamugisha[5], Gote Swedberg[7]

1 Department of Pharmacology, Catholic University of Health and Allied Sciences, Mwanza, Tanzania, 2 Department of Community Medicine, Catholic University of Health and Allied Sciences, Mwanza, Tanzania, 3 National Institute for Medical Research, Tanga Centre, Tanga, Tanzania, 4 Department of Epidemiology, Catholic University of Health and Allied Sciences, Mwanza, Tanzania, 5 Department of Biochemistry, Catholic University of Health and Allied Sciences, Mwanza, Tanzania, 6 Department of Pharmacology and Therapeutics, Makerere University, Kampala, Uganda, 7 Department of Medical Biochemistry and Microbiology, Uppsala University, Uppsala, Sweden

* carol_maro@yahoo.com, karoljuliusmarwa@gmail.com

## Abstract

### Background

Sub-Saharan Africa has the highest burden of malaria in the world. Artemisinin-based combination therapies (ACTs) have been the cornerstone in the efforts to reduce the global burden of malaria. In the effort to facilitate early detection of resistance for artemisinin derivatives and partner drugs, WHO recommends monitoring of ACT's efficacy in the malaria endemic countries. The present systematic meta-analysis study summarises the evidence of therapeutic efficacy of the commonly used artemisinin-based combinations for the treatment of uncomplicated *P. falciparum* malaria in Sub-Saharan Africa after more than a decade since the introduction of the drugs.

### Methods

Fifty two studies carried out from 2010 to 2020 on the efficacy of artemether-lumefantrine or dihydro-artemisinin piperaquine or artesunate amodiaquine in patients with uncomplicated *P. falciparum* malaria in Sub-Saharan Africa were searched for using the Google Scholar, Cochrane Central Register of controlled trials (CENTRAL), PubMed, Medline, LILACS, and EMBASE online data bases. Data was extracted by two independent reviewers. Random analysis effect was performed in STATA 13. Heterogeneity was established using $I^2$ statistics.

**Data Availability Statement:** All relevant data are within the manuscript and Supporting information files.

**Funding:** Authors received no specific fund for this work.

**Competing interests:** The authors have declared that no competing interests exist.

## Results

Based on per protocol analysis, unadjusted cure rates in malaria infected patients treated with artemether-lumefantrine (ALU), artesunate-amodiaquine (ASAQ) and dihydroartemisinin-piperaquine (DHP) were 89%, 94% and 91% respectively. However, the cure rates after PCR correction were 98% for ALU, 99% for ASAQ and 99% for DHP.

## Conclusion

The present meta-analysis reports the overall high malaria treatment success for artemether-lumefantrine, artesunate-amodiaquine and dihydroartemisinin-piperaquine above the WHO threshold value in Sub-Saharan Africa.

## Introduction

Despite the significant progress in malaria reduction since 2010, there is still an estimated 229 million malaria cases occurring worldwide and 409, 000 deaths by 2019 [1]. The malaria case incidence has decreased from 58 in 2015 to 57 in 2019 indicating a decline by 2% while malaria mortality rate reduced from 12 to 10 in the same period [1]. Sub-Saharan Africa harbours a majority of malaria cases with eleven countries accounting for 70% of all the cases and 94% of the recorded deaths [1] Artemisinin-based combination therapies (ACTs) have been the cornerstone in the efforts to reduce the global burden of malaria. However, the gains are jeopardized by the emergence and spread of resistance to artemisinin derivatives and their partner drugs in the Greater Mekong sub-region (GMS) in South-East Asia (SEA).

Artemether-lumefantrine and artesunate-amodiaquine are adopted in treatment guidelines for uncomplicated *p. falciparum* malaria in majority of Sub-Saharan countries while dihydroartemisinin-piperaquine has been introduced in some few countries in the region [1]. Artesunate-mefloquine and artesunate-pyronaridine are not recommended in countries in the region [1]. ACTs that are not recommended by African countries but are available on the market for some Sub-Saharan countries include artesunate-sulfadoxine-pyrimethamine, arterolane-piperaquine, artemisinin-naphthoquine and artemisinin-piperaquine [2, 3].

Resistance has been a driving force for transition of the treatment of falciparum malaria from chloroquine (CQ) to sulphadoxine-pyrimethamine (SP) to artemisinin monotherapy and to the currently WHO recommended artemisinin-based combination therapies (ACTs) [4, 5]. Chloroquine use lasted for about 50 years while SP and artemisinin monotherapy did not last even for a decade [4, 5]. The emergence of resistance to artemisinin derivatives and partner drugs mefloquine, piperaquine and lumefantrine in five countries of the GMS is of great concern to the world. Mutations in K13 propeller region [6] has been associated with delayed parasite clearance in the GMS region. Non-synonymous K13 mutations have been reported in twenty seven Sub-Saharan countries [2]. These mutations are still rare and diverse in Sub-Saharan Africa [2]. The mutation at codon A578S, which is close to C580Y (widely described SNP in SEA), has been frequently reported, however, it was not associated with *in vitro* or *in vivo* resistance [7]. Recently, a mutation at codon R561H was reported in Eastern Rwanda and shown to be associated with delayed parasite clearance [8]. Kelch13 mutants in Africa appear to be indigenous and do not share origin with those in SEA [9].

The Plasmepsin II gene (pfmp2; PFD7 1408000) increased copy number enhances parasite survival under piperaquine (PPQ) exposure through increased aminoacid production to

compensate for the haemoglobin degradation inhibited by PPQ. The pfpm2 multiple copies were detected in Cambodia 2013 and proven to be associated with an increased in vitro piperaquine resistance [10, 11]. The pfmp2 multicopy parasites have also been reported in some parts of Africa including Mali, Tanzania, Uganda, Mozambique, Burkinafaso, Gabon and Ethiopia [12–14] where by isolates from Uganda and Burkinafaso have shown a high frequency of parasites with multiple copies of pfpm2 (>30%) [14]. Mutations in *P.falciparum* chloroquine resistance transporter (pfcrt) and *P.falciparum* exonuclease (pfexo) genes have also been suggested to be associated with PPQ resistance.

In the effort to facilitate early detection of resistance for artemisinin derivatives and partner drugs, WHO recommends monitoring of ACT's efficacy in the malaria endemic countries [1]. Studies done in some parts of Sub-Saharan Africa particularly Kenya, Uganda and Angola a few years after introduction of artemether-lumefantrine (ALU) and dihydroartemisinin-piperaquine (DHP) indicated a decreased rate of parasite clearance and increased recrudescence [15–17] thus posing a great concern since prolonged clearance time (PCT) is the key signal in artemisinin resistance.

In this systematic review and meta-analysis, we summarize the evidence on the efficacy of ACTs used in Sub Saharan Africa from 2010–2020. A recent similar review published while our review was in progress has recorded global estimates for Antimalarial drugs effectiveness from studies done from 1991–2019 [18]. However, our review is different from Rathmes *et al* because our work gives an update on the efficacy for the past ten years only considering there has been some reports on the markers responsible for ACTs resistance in Africa particularly *k-13* and pfmp2 in the recent years thus the efficacy of drugs may change over the years due to resistance or partial resistance. Our review is also specific to Sub-Saharan Africa which is the region accounting for 90% of *P. falciparum* malaria globally where by it is expected drug consumption may be different from other areas. Our review reports the antimalarial drugs efficacy unlike the review by Rathmes *et al* which reports antimalarial drugs effectiveness. Drug effectiveness and efficacy are different study end points/parameters hence findings from the two reviews may not be comparable.

## Methods

### Search strategy

Literature search for published studies assessing the efficacy of Artemether-lumefantrine or artesunate-amodiaquine or dihydroartemisinin-piperaquine from 2010 to 2020 in Sub-Saharan Africa was done using the Cochrane Central Register of Controlled Trials (CENTRAL), EMBASE, Google Scholar, PubMed, Medline and LILACS online data bases.

The search terms used include the following combinations of words: Malaria AND (artemether-lumefantrine OR dihydroartemisinin-piperaquine OR artesunate-amodiaquine) AND (Sub-Saharan Africa) AND efficacy "Dihydroartemisinin piperaquine" OR "Artemether Lumefantrine OR Artesunate Amodiaquine". The search was limited in advanced search to studies conducted for the past ten years because we wanted un update information after more than a decade of ACTs use in Sub-Saharan Africa. The Preferred Reporting Items for Systematic review and Meta-Analysis Protocols (PRISMA-P) 2015 checklist [19] were used to select studies to be included in our review.

### Data extraction

Data extraction was conducted by two independent reviewers. The two reviewers screened the results of the literature search and selected studies to be included in the present study according to the inclusion criteria. Differences in opinion between reviewers on inclusion of studies

were resolved through discussion. Abstracted information /data was entered into extraction sheet which consists of basic and specific information about the studies. The basic information extracted include the author names, country in which the study was carried out, year of study, publication year, years since ACTs introduction, age, sample size, regimen, malaria transmission, study type and baseline characteristics. The specific information include day three parasitaemia, reinfection, recrudescence, Adequate and Clinical Parasitological Response (ACPR), Early Treatment Failure (ETF), Late Clinical Failure (LCF) and Late Parasitological Failure (LPF).

## Inclusion criteria

For the purpose of obtaining recent evidence, all published studies on the efficacy of ACTs in Sub Saharan Africa from 2010 to 2020 were considered for screening. Only studies which recruited subjects from year 2010 were selected. The aim was to have a trend on the efficacy in each country at least after 5 years of clinical use of the drugs as it is known that the efficacy of antimalarial drugs is partly determined by *P.falciparum* resistance which in turn is a function of selection pressure resulting from prolonged use of drugs over time [20].

The primary outcomes were defined as PCR adjusted Adequate and Clinical Parasitological Response (PCR adjusted ACPR) and unadjusted Adequate and Clinical Parasitological Response (PCR-unadjusted ACPR). Secondary outcomes were measurements of recrudescence, re-infection and day 3 parasitaemia.

## Exclusion criteria

We excluded for various reason studies on pregnant women or patients with severe malaria, studies done before 2010, review papers, studies with sample size less than fifty participants, studies on efficacy of ACTs as rescue therapy, studies on ACTs for mass administration or chemoprevention, studies on economic analyses and pharmacokinetics of ACTs, studies which evaluate efficacy of two drugs containing ACTs versus three drug containing combinations, studies that used artemisinin monotherapy, trials assessing safety only, trials comparing three days and five days dosing treatment outcomes, studies which analysed data basing on intention to treat only and studies performed outside Sub-Saharan Africa.

## Methodological and data quality assessment

The national institute of health (NIH) study quality assessment tools for controlled intervention studies and observational cohort and cross sectional studies were used for methodological quality assessment [21]. The score range for the NIH tool scale was from 0 to 14. Each criterion scored one point making up a total of 14 points. The scores were then converted into percentages. The score range of 0–60% was regarded as low quality, 61–80% good quality and 81–100% excellent quality. Any disagreements on extracted data and methodological quality assessment were resolved by consensus between the two independent reviewers. Loss to follow-up was calculated for all studies and was considered as adequate if <10% as per WHO recommendations for antimalarial surveillance studies [22]. Corresponding authors were consulted through email when clarification on data was necessary. All included studies were of good to excellent quality as per the NIH scale shown in Table 1. The possibility of publication bias was assessed by examining asymmetry on funnel plots through STATA.

**Table 1. Characteristics of included studies.**

| SN | Country | Authors | Publication year | Year of Study | study type | ACT | WHO protocol | ACT introducing | Subjects | Age | subjects total | DOF | Score (%) | Ref |
|---|---|---|---|---|---|---|---|---|---|---|---|---|---|---|
| 1 | Kenya | Roth et al. | 2018 | 2015–2017 | Open-label, randomised controlled non-inferiority trial | ALU & PA | Yes | 2006 | CUM | 6moths-12yrs | 96 | 28 & 42 | 100 | [24] |
| 2 | Rwanda | A. Uwimana et al. | 2019 | 2013–2015 | Open label randomised trial | ALU & DHP | Yes | 2005 | CUM | 1–14 yr | 267 | 28& 42 | 93 | [25] |
| 3 | Tanzania | Ishengoma et al. | 2019 | 2016 | Single arm prospective invivo study | ALU | Yes | 2006 | CUM | 6moths-10yrs | 344 | 28 | 79 | [26] |
| 4 | Benin | Ogouyemi-Hounto et al. | 2016 | 2014 | Open-label, non-randomised prospective trial | ALU | Yes | 2004 | CUM | 6months-5 years | 123 | 28 & 42 | 79 | [27] |
| 5 | DR Congo | de Wit et al. | 2016 | 2013 to 2014 | Open label randomised non-inferiority trial | ALU & ASAQ | Yes | 2005 | CUM | 6months-59months | 144 | 28& 42 | 93 | [28] |
| 6 | Ivory Coast | A.Konate et al. | 2018 | 2016 | Controlled randomised open therapeutic trial | ALU & ASAQ | Yes | 2007 | CUM | above 6 months | 120 | 28&42 | 79 | [29] |
| 7 | Mozambique | Salvadoret al. | 2017 | 2015 | Prospective one-arm study | ALU | Yes | 2005 | CUM | 6months-59months | 349 | 28 | 79 | [30] |
| 8 | DR Congo | Singana et al. | 2016 | 2012–2013 | | ALU & ASAQ | Yes | 2005 | CUM | Below 12 yrs | 61 | 28 | 93 | [31] |
| 9 | Niger | Grandesso et al. | 2018 | 2013–2014 | | ALU & DHP | Yes | 2005 | CUM | 6months-59months | 218 | 42 | 93 | [32] |
| 10 | Togo | Dorkenoo et al. | 2016 | 2012–2013 | Prospective study | ALU & ASAQ | Yes | 2005 | CUM | 6months-59months | 261 | 28 | 100 | [33] |
| 11 | Gabon | Ngomo et al. | 2019 | 2014–2015 | Prospective study | ALU & ASAQ | Yes | 2005 | CUM | 12 to 144 months | 106 | 28 | 79 | [34] |
| 12 | Malawi | Paczkowiski et al. | 2016 | 2014 | Randomised invivo efficacy study | ALU & ASAQ | Yes | 2007 | CUM | 6months-59months | 338 | 28 | 93 | [35] |
| 13 | Gabon | Adegite et al. | 2019 | 2017–2018 | Open-label, clinical trial | ALU & ASAQ | Yes | 2013 | CUM | 6months-12 yrs | 50 | 28 | 93 | [36] |
| 14 | Mozambique | Nhama et al. | 2014 | 2011–2012 | Open-label, clinical trial | ALU & ASAQ | Yes | | CUM | 6months-59months | 439 | 28 | 93 | [37] |
| 15 | Tanzania | Kakolwa et al. | 2018 | 2011–2015 | Open-label, one-arm, prospective study | ALU, DHP & ASAQ | YES | 2006 | CUM | 6moths and above | 244 | 28 | 79 | [38] |
| 16 | Tanzania | Mandara et al. | 2018 | 2014–2015 | Open-label, randomised trial | ALU & DHP | Yes | 2006 | CUM | 6months-10 years | 257 | 28,42 and 63 | 100 | [39] |
| 17 | Kenya | Agarwal et al. | 2013 | 2011 | Open-label, invivo trial | ALU & DHP | Yes | 2006 | CUM | 6-59moths | 274 | 28&42 | 79 | [40] |
| 18 | Nigeria | Ebenebe et al. | 2018 | 2014–2015 | Open-label, randomised trial | ALU, ASAQ &DHP | Yes | 2005 | CUM | Below 5 yrs | 992 | 28& 42 | 100 | [41] |
| 19 | Tanzania | Kamugisha etal. | 2012 | 2010–2011 | Prospective single cohort | ALU | Yes | 2006 | CUM | ≤ 5years | 103 | 28 | 86 | [42] |

*(Continued)*

Table 1. (Continued)

| SN | Country | Authors | Publication year | Year of Study | study type | ACT | WHO protocol | ACT introducing | Subjects | Age | subjects total | DOF | Score (%) | Ref |
|---|---|---|---|---|---|---|---|---|---|---|---|---|---|---|
| 20 | Tanzania | Shayo et al. | 2014 | 2013 | Open-label, non-randomised trial | ALU | Yes | 2006 | CUM | 6months-10 years | 88 | 28 | 86 | [43] |
| 21 | Ghana | Abuaku et al | 2012 | 2010–2011 | Prospective study | ALU | Yes | 2008 | CUM | 6months-59months | 175 | 28 | 86 | [44] |
| 22 | Zambia | Ippolito et al. | 2020 | 2014–2015 | Invivo assessment of efficacy | ALU | Yes | 2002 | CUM | 6months-59months | 94 | 28 | 86 | [45] |
| 23 | Ghana | Abuaku et al | 2016 | 2013–2014 | Invivo assessment of efficacy | ALU & ASAQ | Yes | 2008 | CUM | 6months-9 years | 170 | 28 | 79 | [46] |
| 24 | Democratic Republic of Congo | Ndounga et al. | 2015 | 2010–2011 | Randomised trial | ALU &ASAQ | Yes | 2006 | CUM | below 10 yrs | 133 | 28 | 93 | [47] |
| 25 | Democratic Republic of Congo | Onyamboko et al. | 2014 | 2011–2012 | Open-label, randomised controlled trial | ALU, DHP & ASAQ | Yes | 2006 | CUM | 3months-59months | 228 | 28 &42 | 86 | [48] |
| 26 | Mali | Diarra et al. | 2020 | 2015–2016 | Prospective study | ALU & ASAQ | Yes | | CUM | 6months-59months | 225 | 28 & 42 | 93 | [49] |
| 27 | Nigeria | Ojurongbe et al. | 2013 | 2010–2011 | Randomised comparative study | ALU & ASAQ | Yes | 2005 | CUM | 6months-144months | 89 | 28 | 86 | [50] |
| 28 | Sierra Leone | Smith et al. | 2018 | 2015–2016 | Prospective study | ALU, DHP & ASAQ | Yes | 2004 | CUM | 6months-59months | 64 | 28 & 42 | 79 | [51] |
| 29 | Somalia | Warsame et al. | 2019 | 2016–2017 | Single arm, Prospective study | ALU & DHP | Yes | 2006 | CAUM | above 5 years | 139 | 28 & 42 | 93 | [52] |
| 30 | Ethiopia | Ebstie et al. | 2015 | 2012 | Observational cohort | ALU | Yes | 2004 | CAUM | above 5 years | 130 | 28 | 86 | [53] |
| 31 | Tanzania | Mwaiswelo et al. | 2016 | 2014 | Randomised single blinded trial | ALU & ALU plus primaquine | Yes | 2006 | CAUM | 5–23 years | 110 | 28 | 86 | [54] |
| 32 | Ethiopia | Mekonnen et al. | 2015 | 2011 | Invivo therapeutic efficacy | ALU | Yes | 2004 | CAUM | above 6 months | 93 | 28 | 86 | [55] |
| 33 | Senegal | Sylla et al. | 2013 | 2011–2012 | Open randomised trial | ALU, DHP &ASAQ | Yes | 2006 | CAUM | above 6 months | 178 | 28,35 &42 | 79 | [56] |
| 34 | Ethiopia | Abamecha et al. | 2020 | 2017 | Prospective study | ALU | Yes | 2004 | CAUM | above 6 months | 80 | 28 | 86 | [57] |
| 35 | Ethiopia | Wudneh et al. | 2016 | 2014–2015 | Open label invivo trial | ALU | Yes | 2004 | CAUM | above 6 months | 91 | 28 | 86 | [58] |
| 36 | Burkinafaso | Issaka Zongo et al. | 2020 | 2016 | Open randomised controlled trial | ALU & ASAQ | Yes | | CAUM | above 6 months | 138 | 28 | 86 | [59] |
| 37 | Ethiopia | Getnet | 2015 | 2013 | Prospective study | ALU | Yes | 2004 | CAUM | above 6 months | 80 | 28 | 93 | [60] |
| 38 | Mali | Dama et al. | 2018 | 2013–2015 | Randomised open label, controlled trial | ALU & DHP | Yes | 2006 | CAUM | 6months and above | 155 | 28 & 42 | 93 | [61] |
| 39 | Ethiopia | Teklemariam et al. | 2017 | 2014–2015 | Prospective study | ALU | Yes | 2004 | CAUM | $\geq$6 months | 92 | 28 | 86 | [62] |
| 40 | Angola | Kiaco et al. | 2015 | 2011–2013 | Prospective cohort study | ALU | Yes | 2006 | CAUM | > 6 months | 123 | 28 | 86 | [63] |

(Continued)

**Table 1.** (Continued)

| SN | Country | Authors | Publication year | Year of Study | study type | ACT | WHO protocol | ACT introducing | Subjects | Age | subjects total | DOF | Score (%) | Ref |
|---|---|---|---|---|---|---|---|---|---|---|---|---|---|---|
| 41 | Sudan | Adeel et al. | 2016 | 2010–2015 | Prospective study | ALU | Yes | 2004 | CAUM | ≥6 months | 595 | 28 | 93 | [64] |
| 42 | Ethiopia | Deressa et al. | 2017 | 2015–2016 | Prospective study | ALU | Yes | 2004 | CAUM | > 6 months | 80 | 28 | 86 | [65] |
| 43 | Ivory Coast | Yavo et al. | 2015 | 2012 | Open randomised trial | ALU & ASAQ | Yes | 2007 | CAUM | > 2 yrs | 146 | 28 | 79 | [66] |
| 44 | Mali | Niare et al. | 2016 | 2010–2014 | Open label, randomised invivo assay | ALU & AS-SP | Yes | | CAUM | ≥6 months | 237 | 28 | 79 | [67] |
| 45 | Uganda | Muhindo et al. | 2014 | 2011–2012 | Longitudinal randomised controlled trial | ALU &DHP | Yes | | CUM | 4–5 yrs | 202 | 28 | 86 | [68] |
| 46 | Burkina Faso | Sondo et al | 2015 | 2010–2012 | Randomised, open label trial | ALU & ASAQ | Yes | 2005 | CAUM | All age groups | 340 | 28 | 79 | [69] |
| 47 | Ethiopia | Nega et al. | 2016 | 2014–2015 | Open -label trial | ALU | Yes | 2004 | CAUM | ≥6 months | 91 | 28 | 93 | [70] |
| 48 | Sudan | Mohamed et al. | 2017 | 2015–2016 | Open-label clinical trial | DHP&AS-SP | Yes | 2004 | CAUM | > 6 months | 73 | 42 | 86 | [71] |
| 49 | Mauritania | Ouldabdallahi et al. | 2014 | 2013 | Single arm study | ASAQ | Yes | 2006 | CAUM | > 6 months | 130 | 28 | 86 | [72] |
| 50 | Tanzania | Mandara et al. | 2019 | 2017 | Single-arm prospective evaluation | ASAQ& DHP | Yes | 2006 | CUM | 6months-10 yrs | 724 | 28&42 | 93 | [73] |
| 51 | Guinea -Bissau | Ursing et al. | 2016 | 2012–2015 | Randomised, open-label non-inferiority clinical trial | ALU&DHP | Yes | 2008 | CUM | <15 yrs | 157 | 42 | 86 | [74] |
| 52 | Angola | Delvantes et al. | 2018 | 2017 | Invivo assessment of efficacy | ALU&ASAQ&DHP | Yes | 2006 | CUM | > 6 months | 608 | 28&42 | 93 | [75] |

CUM: children with uncomplicated malaria; CAUM: children and adults with uncomplicated malaria; ALU: Artemether Lumefantrine; DHP: Dihydroartemisinin Piperaquine; WHO: World Health Organisation; ACT: Artemisinin Based Combination Therapy; DOF: Number of days of follow up.

## Data collection and analysis

Data were extracted to allow for per-protocol analysis. Meta-analyses were performed using STATA 13 (Statistical Corporation, College Station, TX, US). Random effects model was used to combine information from comparable studies. The heterogeneity between studies was evaluated using Cochran's Q and $I^2$. Heterogeneity was considered substantial when p-value of Q was <0.10 and or /$I^2$ was >50% [23].

# Results

## Study characteristics

A total of 2,639 records (after removal of duplications) were identified through the electronic data base search as shown in Fig 1. Eighty two articles were included for full-text review. A total of 52 studies were eligible for data extraction, according to the inclusion criteria. These studies originated from 25 countries of Sub-Saharan Africa. Most studies (70%) were done at least 9 years after the introduction of ACT use in the respective countries. All studies were carried out according to the WHO standardized Protocol (2003 or 2009) for monitoring anti-malarial drug efficacy. Thirty two studies enrolled children only where as 20 studies enrolled children and adults as study participants. A total of 11,053 subjects were enrolled in the studies where by the number of subjects ranged from 50 to 992. The details on the study characteristics are indicated in Table 1. The treatment groups for the studies were as follows: ALU& DHP (n = 7), ALU&ASAQ(n = 17), ALU&ASAQ& DHP(n = 5), ALU&PA(n = 1), ALUU&AS-SP (n = 2), DHP & ASAQ(n = 1), ASAQ(n = 1) only, ALU(n = 17) and ALU&ALU plus primaquine(n = 1). In studies involving ALU and ASAQ, patients were followed up to 28 days. However, in all studies involving DHP patients were followed up to 42 days due to the long half-life of piperaquine. Early treatment failure was reported in ten studies (for ALU), two studies (for ASAQ) and three studies (for DHP). PCR Unadjusted cure rates below 90% were recorded in nineteen, seven and four studies for ALU, ASAQ and DHP respectively. PCR adjusted cure rates below 90% were reported in one, one and zero studies for ALU, ASAQ and DHP respectively.

## Baseline characteristics of the subjects

A total of 11,053 patients with uncomplicated *P. falciparum* malaria were included in the meta-analysis. The mean age ranged between 30.0 and 268.0 months old. The male's proportion was 52.6%. The mean axillary temperature at day 0 ranged between 37 and 39.2 centigrade. The mean Hemoglobin (g/dl) at day zero also ranged between 8.9 and 13.7. At recruitment, the average parasite count per patient was 4,473–51,300.

## Artemether-lumefantrine (ALU)

A meta-analysis was conducted for forty six studies to explore the overall treatment outcomes in Sub-Saharan Africa. Based on per protocol analysis, day 28 unadjusted cure rate was low (89%) (Fig 2). However, the day 28 cure rate was 98% after PCR correction (Fig 3). The recrudescence and reinfection rates after 28 days were 2% and 10% respectively (S1 and S4 Figs). Only 1% of the children had parasitaemia on day 3. Early treatment failure was only observed in less than 0.2% of the patients.

## Artesunate-amodiaquine (ASAQ)

Twenty four studies were included in the meta-analysis to explore the overall treatment outcomes in Sub-Saharan Africa. Based on per protocol analysis, day 28 unadjusted cure rate was

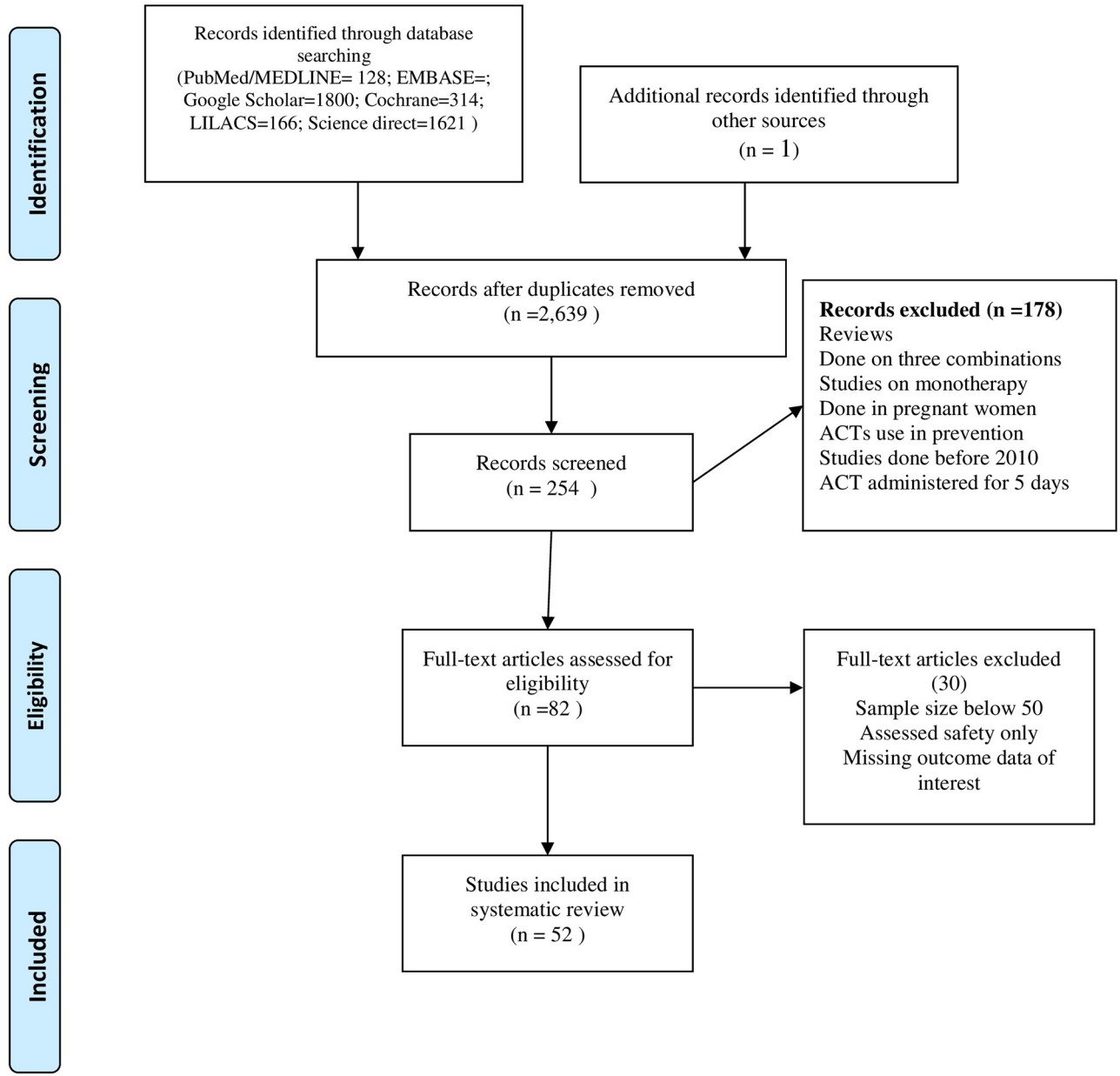

**Fig 1. PRISMA flow diagram for article search and screening.**

high (94%) (Fig 4). The day 28 cure rate was 99% after PCR correction (Fig 4). The recrudescence and reinfection rates after 28 days were 1% and 4% respectively (S2 and S5 Figs). Only less than 1% of the children had parasitaemia on day 3. Early treatment failure was observed in less than 0.1% of the patients.

## Dihydroartemisinin-piperaquine (DHA-PPQ)

Fifteen studies were included in meta-analysis. Based on per protocol analysis, day 42 unadjusted cure rate was 91% (Fig 5). However, the day 42 cure rate was 99% after PCR correction

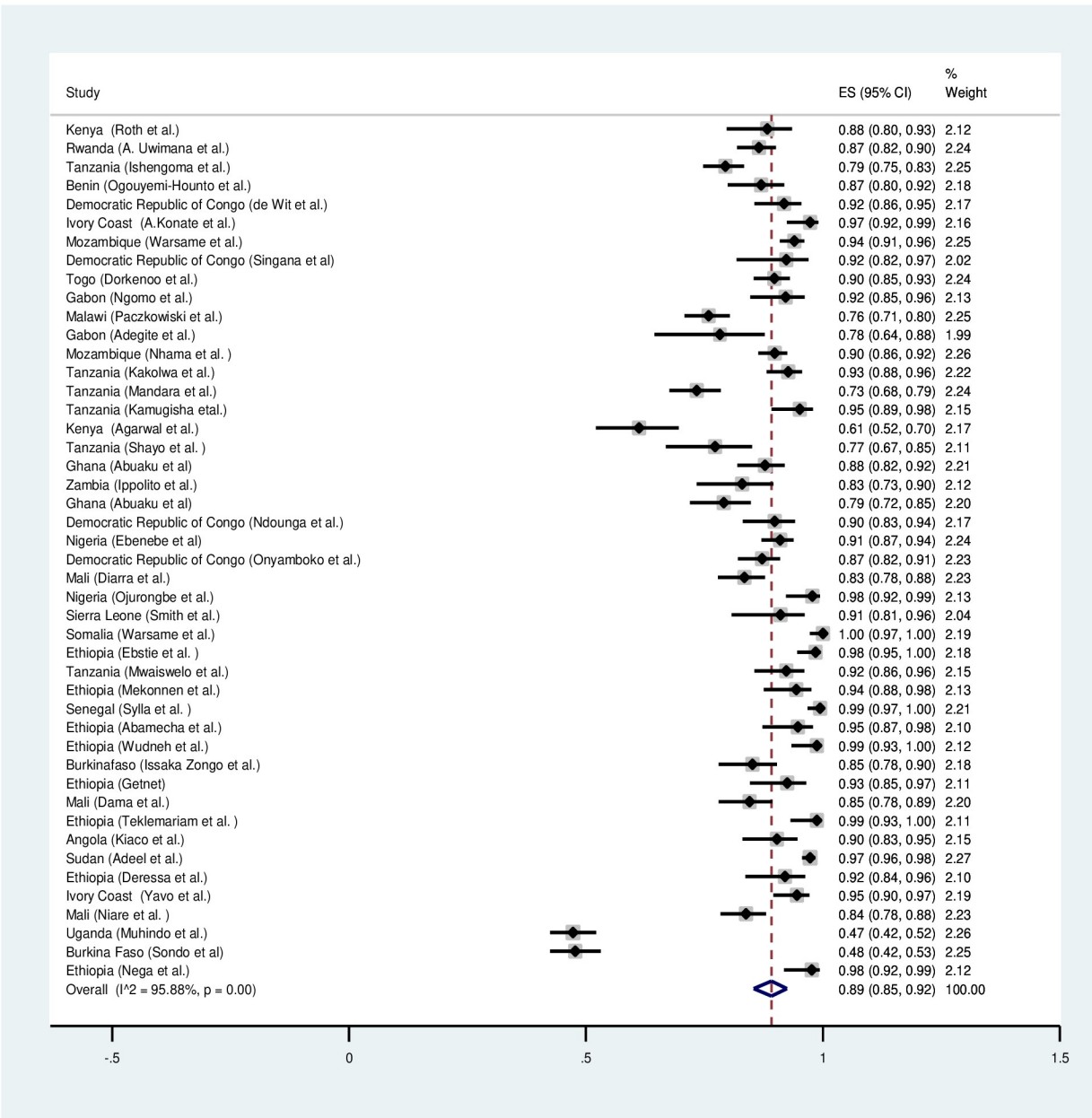

**Fig 2. Forest plot for artemether-lumefantrine PCR unadjusted cure rate based on the per protocol analysis.**

(Fig 5). The recrudescence and reinfection rates were <0.5% and 5% respectively (S3 and S6 Figs). Less than 1% of the children had parasitaemia on day 3. Early treatment failure was observed in less than 0.3% of the patients.

## Discussion

The present metanalysis shows that the ACTs evaluated are still efficacious with PCR corrected efficacies greater than 90% which is the WHO minimum threshold requirement for recommending of a change in the treatment policy [1, 2]. Early treatment failure did not exceed 0.4%

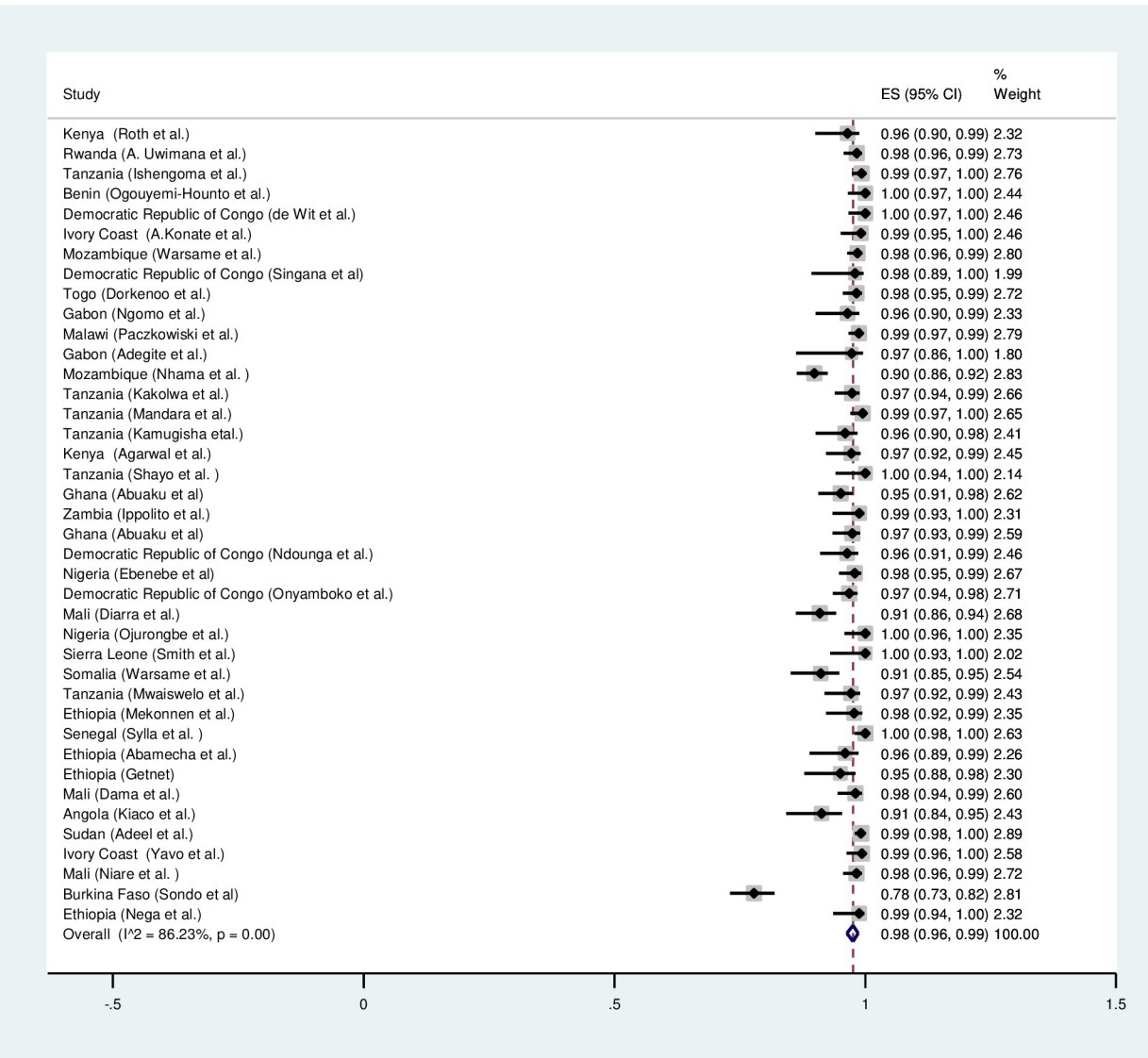

**Fig 3. Forest plot for artemether-lumefantrine PCR adjusted cure rate based on the per protocol analysis.**

in ALU, ASAQ or DHP. All ACTs studied have recorded a rapid parasite clearance equal or above 99% on day 3. These drugs have retained high efficacy (PCR corrected cure rate) in the treatment of uncomplicated *P.falciparum* malaria after more than a decade since the introduction of ACTs in Sub Saharan Africa. Recrudescence was low in general but higher for ALU (2%) compared to ASAQ (1%) and DHP (<0.5%). Although our meta-analysis confirms that the three ACTs have retained high efficacy in the Sub-Saharan region, it does, however demonstrate a high re-infection rate for ALU (10%).

The retained high efficacy of the ACTs studied may be due to the following reasons: *Plasmodium falciparum* Kelch 13 (pf K13) mutations exist generally at a low frequency in Africa and there is no evidence of the mutation's association with slow clearing parasites in the region [26, 76] with an exception of some parts of Rwanda [77]. Parasites in Africa seem to be under less evolutionary pressure to develop ACT resistance compared to those found in South East

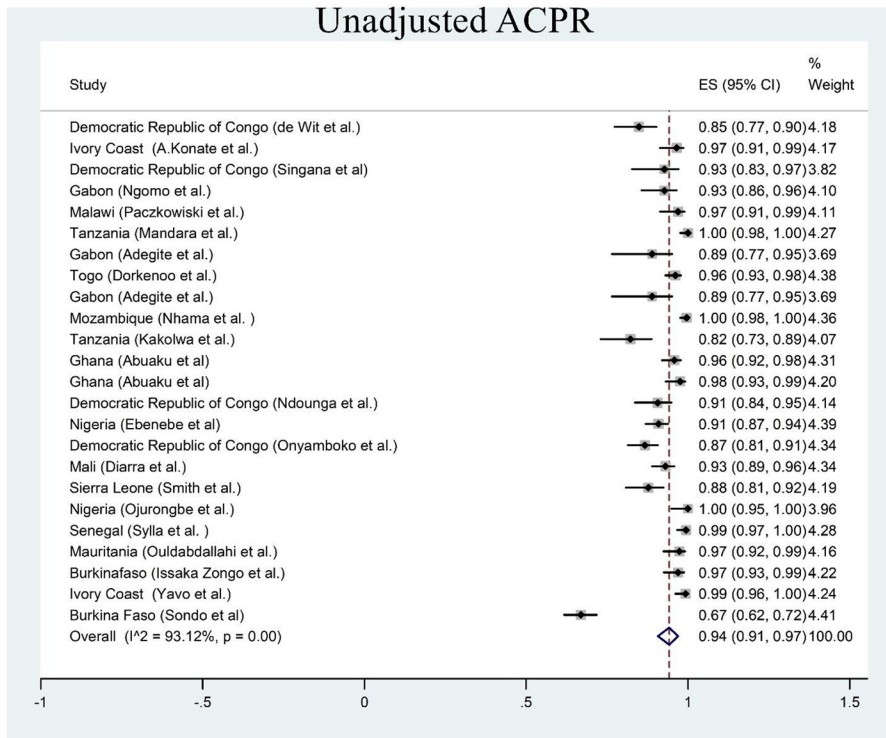

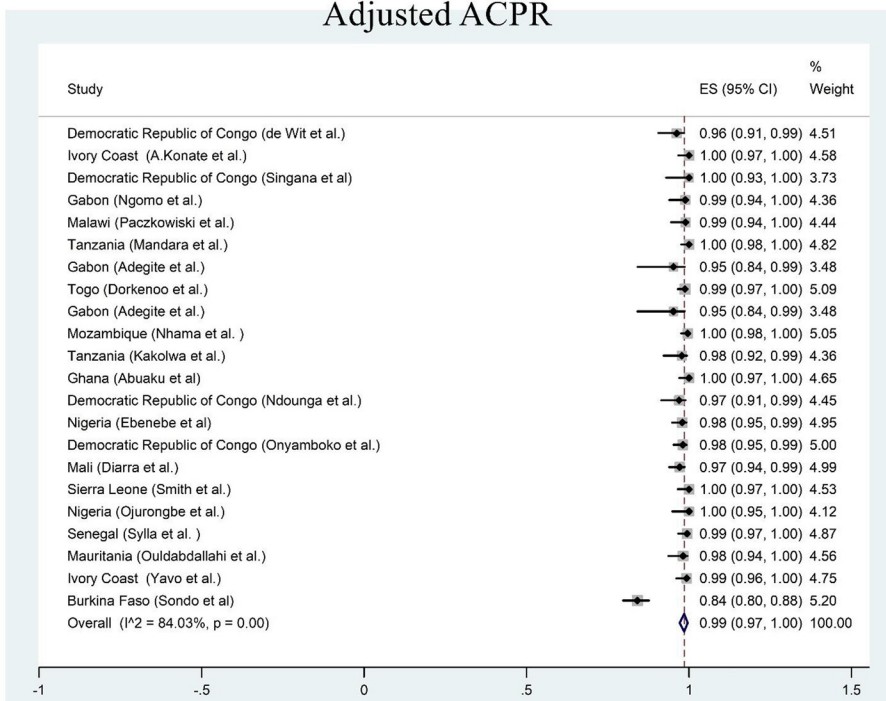

**Fig 4. Forest plot for artesunate-amodiaquine PCR unadjusted and adjusted cure rate based on the per protocol analysis.**

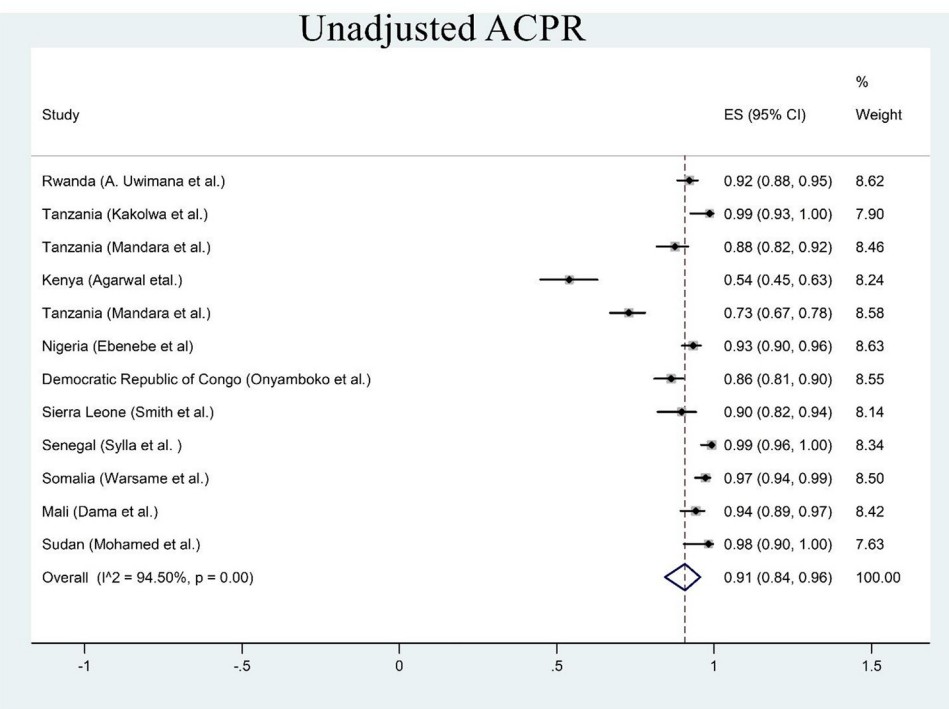

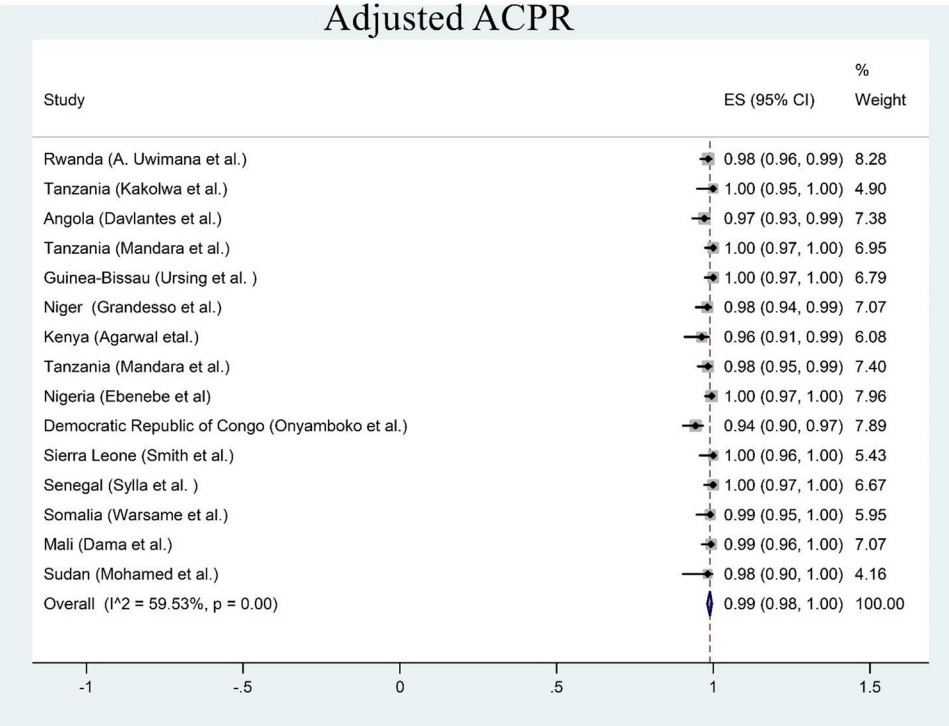

**Fig 5. Forest plot for dihydroartemisinin-piperaquine PCR unadjusted and adjusted cure rate based on the per protocol analysis.**

Asia region where artemisinin was widely used as monotherapy before adoption of ACTs [26]. Individuals in endemic settings in Africa have frequent exposure to parasites due to the high malaria endemicity, hence the high level of naturally acquired immunity often lead to asymptomatic infections. These are less exposed to drug pressure from antimalarial treatment as they are not treated. This in turn may account for the lack of slow clearing parasites and the low frequency of the *pf K13* mutations. High level of acquired immunity in the studied region where malaria is endemic may also be a possible explanation for the documented high efficacy in our review. Another factor could be low level occurrence of parasite background mutations (fdmdr2, arps10) in the parasite genome in contrast to parasite population in SEA.

The observed high level of re-infection in patients treated with ALU does not necessarily indicate failing of the drug. However, it does indicate that malaria episodes are common in patients treated with ALU in the region. This poses a public health concern and economic burden to the health systems and community. The documented high re-infection rate among patients treated with ALU may be attributed to the less post-treatment protection of lumefantrine compared to other partner drugs with longer half-lives (piperaquine and amodiaquine). Lumefantrine use quite rapidly selects for mutants with lower susceptibility to lumefantrine, and thus the protective concentration for lumefantrine is increased, which in turn leads to a shorter time for the protection effect. Other mutations in variable genes like pfupb-1 and pfap2mu, may have a role in the shorter protection and thus making reinfections more likely with ALU.

High ALU unadjusted total treatment failure/low unadjusted ACPR and recrudescence is alarming. Firstly, because uncorrected parasitaemia in form of microscopy blood smears is the main tool used to make clinical decision whether there is cure, clinical resistance or a need for switch therapy [20]. The high ALU uncorrected ACPR may be due to the limitations of microscopy in detecting parasites compared to PCR suggesting a need for employing PCR as diagnostic method in the region. Secondly, recrudescent infections tend to stimulate the production of gametocytes which in turn tend to facilitate the transmission of resistance. Thirdly, ALU is used as first line anti-malarial for the treatment of *P. falciparum* uncomplicated malaria in most malaria endemic countries in the WHO Africa region [1].

Meta-analysis on PCR adjusted day 28 total treatment success indicates ASAQ is as efficacious as ALU and DHP in Sub-Saharan Africa. ASAQ has shown the highest PCR unadjusted efficacy than both ALU and DHP (94 vs 89 and 91 respectively). ASAQ has retained high efficacy possibly due to its limited use owing to clinicians preferring to prescribe ALU over ASAQ avoiding high risk of neurological side effects associated with ASAQ. This has led some African countries to omit ASAQ from their treatment guidelines [20]. It is also possible ASAQ is benefiting from the *P. falciparum* revision to chloroquine sensitivity as documented recently in different parts of Sub-Saharan Africa. The revision to parasites with wild type *pfmdr1* and *pfcrt* alleles sensitive to chloroquine and amodiaquine [78–80] is a great advantage to ASAQ.

The overall day 42 PCR adjusted efficacy for DHP was similar to ALU and ASAQ. DHP also has recorded a lower re-infection rate similar to ASAQ but much less than ALU. The contribution of the partner drug piperaquine to the parasite killing effect soon after drug administration may account for the high efficacy observed [41]. Piperaquine has a longer half-life (2–3 weeks) than lumefantrine (4.5 days) which may also explain the lower re-infection rate observed with DHP than ALU. It is also possible that *P. falciparum* isolates in Sub Saharan Africa have a high sensitivity to piperaquine. This argument can be supported by the evidence that pfmp2 multicopies have not been reported to be associated with treatment failure or delayed parasite clearance in Africa unlike in SEA. DHP in most Sub-Saharan countries is not deployed as first line but second line or alternative treatment. The use of DHP is limited owing

to the high cost as the drug is not subsidised in most African countries unlike ALU, and this may account for less drug pressure on *P. falciparum* and hence the high efficacy is retained.

The newly documented use of DHP (piperaquine being an aminoquinoline) as IPT in pregnancy and mass administration as prophylaxis in some Sub Saharan countries is not associated with selection of the pcrt and pfmdr1 mutations observed with the use of chloroquine and other aminoquinolines [13]. However, in some few parts of Africa the use of DHP as chemoprevention is associated with selection of parasites associated with resistance to aminoquinolines [13, 81]. In Cambodia, DHP as IPT in pregnancy is observed not to select for multicopy pfmp2 parasites. Mutations in other genes accounting for piperaquine resistance are less frequent in Africa than South East Asia. Generally, it is not clear what impact chemoprevention practice have on the selection for *P. falciparum* resistance and the efficacy of DHP in future.

In general, the efficacies recorded in this metanalysis are comparable with those from metanalyses done in Africa before 2010 [82, 83] suggesting that the drugs have retained efficacies after more than a decade since introduction. The reasons discussed above may account for these drugs retaining their efficacy over the past years. A recent similar review published while our review was in progress has recorded global estimates for Antimalarial drugs effectiveness from studies done from 1991–2019 [18]. The review has reported global estimation of ACT effectiveness below 72% from 2016–2019. The present review reports the efficacy of ALU, DHP and ASAQ from 2010–2020. The findings from our review cannot be compared to the review by Rathmes *et al* due to the differences in the primary end points where by we report drug efficacy unlike the other review which reports drug effectiveness.

This review has some limitations. Not all countries have been represented in this review due to our inclusion criteria. Our review considered only treatment outcomes data as per protocol analysis, the intention to treat treatment outcomes was not considered. The present metanalysis did not evaluate the safety of the ACTs studied because this has been extensively reviewed elsewhere. Our review has included only studies conducted from 2010–2020, we understand there is a possibility there could be some studies conducted from the stated period but have delayed to be published hence not included in our review.

## Conclusion

The present meta-analysis reports the overall high malaria treatment success for artemether-lumefantrine, artesunate-amodiaquine and dihydroartemisinin-piperaquine above the WHO threshold value suggesting there is no need for a change in treatment policy in Sub-Saharan countries. However, there is a need for intensifying the monitoring of molecular makers for resistance of artemisinin derivatives and their partner drugs. The documented high reinfection rate with ALU calls for intensification of malaria prevention interventions in the region.

## Supporting information

**S1 Fig. Recrudescence for artemether-lumefantrine.**
(DOCX)

**S2 Fig. Recrudescence for artesunate-amodiaquine.**
(DOCX)

**S3 Fig. Recrudescence for dihydro-artemisinin piperaquine.**
(DOCX)

**S4 Fig. Reinfection for artemether-lumefantrine.**
(DOCX)

**S5 Fig. Reinfection for artesunate-amodiaquine.**
(DOCX)

**S6 Fig. Reinfection for dihydroartemisinin-piperaquine.** Abbreviations: ALU:artemether-lumefantrine; DHP:dihydroartemisinin-piperaquine; ASAQ:artesunate-amodiaquine; WHO: World Health Organization; PCR:polymerase chain reaction.
(DOCX)

**S1 Checklist.**
(DOC)

**S1 Data.**
(DTA)

**S2 Data.**
(DTA)

**S3 Data.**
(DTA)

## Author Contributions

**Conceptualization:** Karol Marwa, Anthony Kapesa, Erasmus Kamugisha, Gote Swedberg.

**Data curation:** Karol Marwa, Anthony Kapesa.

**Formal analysis:** Karol Marwa, Evelyne Konje, Benson Kidenya.

**Methodology:** Karol Marwa, Vito Baraka, Benson Kidenya.

**Software:** Evelyne Konje, Benson Kidenya.

**Supervision:** Vito Baraka, Jackson Mukonzo, Erasmus Kamugisha, Gote Swedberg.

**Validation:** Evelyne Konje, Benson Kidenya, Jackson Mukonzo.

**Writing – original draft:** Karol Marwa.

**Writing – review & editing:** Karol Marwa, Anthony Kapesa, Vito Baraka, Jackson Mukonzo, Erasmus Kamugisha, Gote Swedberg.

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
