## [Decision Letter · Decision Letter 0]

13 Aug 2021

PONE-D-21-19495

Therapeutic efficacy of Artemether-Lumefantrine, Artesunate-Amodiaquine and Dihydroartemisinin -Piperaquine in the treatment of uncomplicated malaria in Sub-Saharan Africa. A decade after the introduction: a systematic review and meta-analysis

PLOS ONE

Dear Dr. Marwa,

Thank you for submitting your manuscript to PLOS ONE. After careful consideration, we feel that it has merit but does not fully meet PLOS ONE’s publication criteria as it currently stands. Therefore, we invite you to submit a revised version of the manuscript that addresses the points raised during the review process.

In addition to addressing reviewer comments, please be sure to reference other similar recent work and compare your findings. For example the similar meta-analysis by Rathmes et al Mal J 2020 Global estimation of anti-malarial drug effectiveness for the treatment of uncomplicated *Plasmodium falciparum* malaria 1991–2019. https://malariajournal.biomedcentral.com/articles/10.1186/s12936-020-03446-8 

We look forward to receiving your revised manuscript.

Kind regards,

Lucy C. Okell

Academic Editor

PLOS ONE

Journal Requirements:

2. As noted previously by Academic Editor, your article is similar to the following publication:

https://malariajournal.biomedcentral.com/articles/10.1186/s12936-020-03446-8

Please cite and discuss the above study in the introduction and discussion sections of your manuscript, clarifying how the present work is related to the previously published paper.

Please note that our second publication criterion states that "If a submitted study replicates or is very similar to previous work, authors must provide a sound scientific rationale for the submitted work and clearly reference and discuss the existing literature. Submissions that replicate or are derivative of existing work will likely be rejected if authors do not provide adequate justification." http://www.plosone.org/static/publication.action#results.

Thank you for your attention to this request.

5. Please upload a copy of Figure 8, to which you refer in your text on page 15. If the figure is no longer to be included as part of the submission please remove all reference to it within the text.

Reviewers' comments:

Reviewer's Responses to Questions

**Comments to the Author**

1. Is the manuscript technically sound, and do the data support the conclusions?

Reviewer #1: Yes

Reviewer #2: Partly

2. Has the statistical analysis been performed appropriately and rigorously? 

Reviewer #1: I Don't Know

Reviewer #2: I Don't Know

3. Have the authors made all data underlying the findings in their manuscript fully available?

Reviewer #1: Yes

Reviewer #2: Yes

4. Is the manuscript presented in an intelligible fashion and written in standard English?

Reviewer #1: Yes

Reviewer #2: Yes

5. Review Comments to the Author

Reviewer #1: Comments and suggestions:

The scope of the manuscript is relevant. Unfortunately, it has major deficiencies including gaps in the adequacy/correctness of publicly available facts/information as listed below:

• Title: 'Therapeutic efficacy of Artemether-Lumefantrine, Artesunate-Amodiaquine and Dihydroartemisinin -Piperaquine in the treatment of uncomplicated malaria in Sub-Saharan Africa. A decade after the introduction: a systematic review and meta-analysis". The phrase "A decade after the introduction" applies to AL and ASAQ, but not to DHA -PPQ, which has only recently been recommended in Africa. Therefore, the authors should revise the title.

• Often, efficacy studies are published several years after they were conducted. Therefore, not all studies conducted between 2010-2020 are included in this review. This should be stated in the study limitation.

• "P. falciparum" should always be "P. falciparum" in italics. The authors should correct this in the manuscript.

Lines: 58-60: Authors should use the most recent malaria burden data from the World Malaria Report 2020.

• Lines 61-64: These statements need reference/s.

• Line 63: change "partnerdrugs" to "partner drugs".

• Lines 65-66: the reference cited by the authors for the statement "Artemether-lumefantrine and artesunate-amodiaquine are the currently most commonly employed ACTs in 66 Sub-Saharan Africa [1]" is not appropriate. The reference is about the updated treatment guidelines that include ACTs and NOT, which ACTs are most commonly recommended. The correct reference is World Malaria report 2020 (see ANNEX 3 - B. ANTIMALARIAL DRUG POLICY, 2019).

• Line 67: "Artesunate-mefloquine and artesunate-pyronaridine are not used in most countries in the region." Should be revised as "Artesunate-mefloquine and artesunate-pyronaridine are not recommended in countries in the Region."

• Lines 68-70: "ACTs not endorsed by WHO for Africa but are available in the market for some Sub-Saharan countries include artesunate-sulfadoxine-pyrimethamine, arterolane-piperaquine, artemisinin-naphthoquine, and artemisinin-piperaquine [1, 3]." As clearly stated in the WHO treatment guidelines, WHO provides global guidelines and does not dictate which ACTs are recommended at the country level. It is the countries that decide which ACT/s they prefer. Artesunate-sulfadoxine-pyrimethamine has been used in African countries (Somalia and Sudan), although it has recently been abandoned. Therefore, the authors need to revise the statement, "ACTs that are not recommended by African countries (wmr 2020) but are available on the market for some Sub-Saharan countries include arterolane-piperaquine, artemisinin-naphthoquine, and artemisinin-piperaquine [3]."

• Lines 81-83: "Recently, a de novo mutation at codon R561H was reported in eastern Rwanda, although it was not linked with delayed parasite clearance in vivo but gene editing demonstrated its potential to drive in vitro artemisinin resistance." The statement needs a reference. Also, the authors should be aware that a subsequent study in Rwanda showed a higher number of R561H mutations and an association between the mutation and delayed parasite clearance.

• Lines 88-89: "The pfmp2 multicopy parasites have also been reported in some parts of Africa, including Mali, Tanzania, Uganda, and Ethiopia [13, 14]." There are other studies that detected pfmp2 multicopy parasites in Africa: A study by Leroy et al 2019, with 2014/2015 samples collected from Benin, Burkina Faso, DRC, Mozambique and Uganda reported multicopy Pfpm2 varying from 11.3% to 33.9%. The authors need to conduct a proper literature search on this topic.

• Line 92-93: Statement "In the effort to facilitate early detection of resistance for artemisinin derivatives and partner drugs, WHO recommends monitoring of ACT's efficacy in the malaria-endemic countries [15]. This document has recently been updated and authors should use the new version: "WHO. Report on antimalarial drug efficacy, resistance and response: 10 years of surveillance (2010-2019). Geneva: World Health Organization; 2020b. https://www.who.int/publications/i/item/9789240012813.

• Lines 98-99: "Most studies to assess the efficacy of ACTs in Africa were conducted between 2005 and 2009, just few years post the introduction of the drugs." This is incorrect. There are many efficacy studies that were conducted after 2009.

• Lines 99-100: "In this systematic review and meta-analysis, we summarize the evidence on the efficacy of ACTs used in Sub Saharan Africa for the past ten years." It would be better to provide reference points: from which year to which year. Suggest: "In this systematic review and meta-analysis, we summarize the evidence on the efficacy of ACTs used in sub-Saharan Africa from 2010 to 2020."

• Lines 234-236: "The present metanalysis shows that the ACTs evaluated are still efficacious with PCR corrected efficacies greater than 95% which is the WHO minimum threshold requirement for recommendation of a change in the treatment policy [7, 22]." WHO recommends treatment policy change when the efficacy of ACT falls below 90%. The 95% threshold applies to newly introduced first-line treatments. Therefore, the statement should be revised to read, "The present meta-analysis shows that the ACTs evaluated are still efficacious with PCR-corrected efficacies greater than 90%, which is the WHO minimum threshold requirement for recommending a change in treatment policy [WHO. Report on antimalarial drug efficacy, resistance and response: 10 years of surveillance (2010-2019)]."

• In the tables: the authors need to correct the first author of the Mozambique study published in 2017. It is Salvador et al. 2017 as in the reference list, not Warsame M.

• References:

o The authors have used different styles for the author of WHO: e.g Organization WH. and WHO. Some of these references are not complete. The author should follow the Malaria Journal reference style.

o The authors used outdated WHO documents, although updated versions are available:

o World Malaria Report 2015 (refs 7 and 2) and World Malaria Report 2019 are used. Suggest to use World Malaria Report 2020.

o Ref 8: "WHO. Artemisinin and artemisinin-based combination therapy resistance: status - report. World Health Organization, 2016." There are many updates to this series after this version. Authors must use the latest update. I would suggest that the authors use the updated document on the topic "WHO. Report on antimalarial drug efficacy, resistance and response: 10 years of surveillance (2010-2019). Geneva: World Health Organization; 2020b. " ext-link-type="uri" xlink:type="simple">https://www.who.int/publications/i/item/9789240012813."

o Ref 15: "Organization WH. Global report on antimalarial drug efficacy and drug resistance: 2000-2010. The reference is out of date and also does not conform to the MJ reference style. Authors must use the new updated version "WHO. Report on antimalarial drug efficacy, resistance and response: 10 years of surveillance (2010-2019). Geneva: World Health Organization; 2020b. " ext-link-type="uri" xlink:type="simple">https://www.who.int/publications/i/item/9789240012813."

o Ref 22: "Organization WH. Methods for surveillance of antimalarial drug efficacy. 2009." Is not consistent with MJ style. All referenced WHO documents in the manuscript should be carefuly revised.

o Ref #5 as listed in the manuscript is wrong: Njagi EN, Orinda GO, Thiongo K, Kimani FT, Matoke-Muhia D. Clinical efficacy of artemisininlumefantrine and status of antifolate drug resistance markers in western Kenya. 2019. 

Correct Ref #5: Kishoyian G, Njagi ENM, Orinda GO, Kimani FT, Thiongo K, Matoke-Muhia D. Efficacy of artemisinin-lumefantrine for treatment of uncomplicated malaria after more than a decade of its use in Kenya. Epidemiol Infect. 2021 Jan 5;149:e27. doi: 10.1017/S0950268820003167.

Reviewer #2: The manuscript entitled “Therapeutic efficacy of Artemether-Lumefantrine, Artesunate-Amodiaquine and Dihydroartemisinin -Piperaquine in the treatment of uncomplicated malaria in Sub Saharan Africa. A decade after the introduction: a systematic review and meta-analysis” (PONE-D-21-19495) is well written. In this manuscript the authors discussed one of the important issues. While I have no doubt that this was a very interesting study and well planned. The followings are the concern of the MS.

1. The study for all the drug was conducted in Sub Saharan Africa and the number of studies for each drug may be presented in a table form with more cohesive manner as provided in line no 176-178 is very confusing.

2. Table no. 1 is need to reframe as it is not in presentable form.

3. Author should provide the others factor responsible for high difference in efficacy among PCR corrected vs uncorrected case and also the transmission aspect may be discussed.

4. The above finding clearly indicated that there is need to monitor the molecular markers of both the drugs and the policy makers may consider this as on priority.

5. I believe that parasite clearance time should be tabled as it is important findings.

1. I feel that the concluding message from author based on the review and meta-analysis is need more attention for the policy makers as well as the other stakeholders as author fail to discuss the importance of the results of the study.

6. The language is mostly suitable for publication; however, the entire article would benefit from a careful review to eliminate some few grammatical and spelling errors.

6. PLOS authors have the option to publish the peer review history of their article (what does this mean?). If published, this will include your full peer review and any attached files.

Reviewer #1: No

Reviewer #2: No

---

## [Author Response · Author response to Decision Letter 0]

9 Nov 2021

Dear editor,

I will like to thank the editors and reviewers for their comments/criticism which have made the manuscript (PONE-D-21-02779) much better. Please receive the revised manuscript. 

Reviewers comments have been addressed as shown in the table below and the manuscript with track changes. 

SNo. Reviewer’s 1 comment Response 

1 The phrase "A decade after the introduction" applies to AL and ASAQ, but not to DHA -PPQ, which has only recently been recommended in Africa. Therefore, the authors should revise the title Authors meant a decade after introduction of artemisinin-based combination therapies. The title has been modified to clarify what authors meant. 

2 Often, efficacy studies are published several years after they were conducted. Therefore, not all studies conducted between 2010-2020 are included in this review. This should be stated in the study limitation Reviewer’s comments have been included in the limitation as shown in the manuscript

3 "P. falciparum" should always be "P. falciparum" in italics. The authors should correct this in the manuscript. Corrections have been made as suggested by the reviewer

4 

3 Line 63: change "partnerdrugs" to "partner drugs". Editing has been done as suggested by the reviewer 

4 Lines 65-66: the reference cited by the authors for the statement "Artemether-lumefantrine and artesunate-amodiaquine are the currently most commonly employed ACTs in 66 Sub-Saharan Africa [1]" is not appropriate. A new reference has been inserted as suggested by the reviewer 

5 Line 67: "Artesunate-mefloquine and artesunate-pyronaridine are not used in most countries in the region." Should be revised as "Artesunate-mefloquine and artesunate-pyronaridine are not recommended in countries in the Region." The statement has been revised as suggested by the reviewer 

6 Therefore, the authors need to revise the statement, "ACTs that are not recommended by African countries (wmr 2020) but are available on the market for some Sub-Saharan countries include arterolane-piperaquine, artemisinin-naphthoquine, and artemisinin-piperaquine [3]." We have revised the statement as suggested by the reviewer

7 Line 81-83." The statement needs a reference. Also, the authors should be aware that a subsequent study in Rwanda showed a higher number of R561H mutations and an association between the mutation and delayed parasite clearance. A reference has been inserted. The sentence has been re-written as seen in the manuscript

8 Lines 88-89: There are other studies that detected pfmp2 multicopy parasites in Africa: A study by Leroy et al 2019, with 2014/2015 samples collected from Benin, Burkina Faso, DRC, Mozambique and Uganda reported multicopy Pfpm2 varying from 11.3% to 33.9%. The authors need to conduct a proper literature search on this topic. The sentence has been re-written as seen in the manuscript and new references have been inserted. 

9 • Line 92-93: This document has recently been updated and authors should use the new version: "WHO. Report on antimalarial drug efficacy, resistance and response: 10 years of surveillance (2010-2019). A new version of the document has now been cited as suggested by the reviewer 

11 • Lines 98-99: "Most studies to assess the efficacy of ACTs in Africa were conducted between 2005 and 2009, just few years post the introduction of the drugs." This is incorrect. The sentence has been omitted 

12 Lines 99-100: It would be better to provide reference points: from which year to which year. Suggest: "In this systematic review and meta-analysis, we summarize the evidence on the efficacy of ACTs used in sub-Saharan Africa from 2010 to 2020." We have revised the sentence as suggested by the reviewer

13 • Lines 234-236: Therefore, the statement should be revised to read, "The present meta-analysis shows that the ACTs evaluated are still efficacious with PCR-corrected efficacies greater than 90%, which is the WHO minimum threshold requirement for recommending a change in treatment The statement has been revised and a new reference cited as suggested by the reviewer

14 • In the tables: the authors need to correct the first author of the Mozambique study published in 2017. It is Salvador et al. 2017 as in the reference list, not Warsame M. The first author name has now been corrected

15. The authors have used different styles for the author of WHO: e.g Organization WH. and WHO. Some of these references are not complete. The author should follow the Malaria Journal reference style. We seek editor’s guidance on this because our references are in accordance to PLOS style

1. The authors used outdated WHO documents, although updated versions are available:

o World Malaria Report 2015 (refs 7 and 2) and World Malaria Report 2019 are used. Suggest to use World Malaria Report 2020. The update reference has been inserted as suggested by the reviewer

2. Ref 8: "WHO. Artemisinin and artemisinin-based combination therapy resistance: status - report. World Health Organization, 2016." There are many updates to this series after this version. Authors must use the latest update. I would suggest that the authors use the updated document on the topic "WHO. Report on antimalarial drug efficacy, resistance and response: 10 years of surveillance (2010-2019). Geneva: World Health Organization; 2020b. https://www.who.int/publications/i/item/9789240012813." The update reference has been inserted as suggested by the reviewer

3. Ref 15: "Organization WH. Global report on antimalarial drug efficacy and drug resistance: 2000-2010. The reference is out of date and also does not conform to the MJ reference style. Authors must use the new updated version "WHO. Report on antimalarial drug efficacy, resistance and response: 10 years of surveillance (2010-2019). Geneva: World Health Organization; 2020b. https://www.who.int/publications/i/item/9789240012813." The update reference has been inserted as suggested by the reviewer

4. o Ref 22: "Organization WH. Methods for surveillance of antimalarial drug efficacy. 2009." Is not consistent with MJ style. All referenced WHO documents in the manuscript should be carefuly revised. Editing has been made to meet PLOS style and not MJ

5. Ref #5 as listed in the manuscript is wrong: Njagi EN, Orinda GO, Thiongo K, Kimani FT, Matoke-Muhia D. Clinical efficacy of artemisininlumefantrine and status of antifolate drug resistance markers in western Kenya. 2019. 

Correct Ref #5: Kishoyian G, Njagi ENM, Orinda GO, Kimani FT, Thiongo K, Matoke-Muhia D. Efficacy of artemisinin-lumefantrine for treatment of uncomplicated malaria after more than a decade of its use in Kenya. Epidemiol Infect. 2021 Jan 5;149:e27. doi: 10.1017/S0950268820003167. The errors have been corrected. The correct reference has now been inserted

 Reviewer 2 

1. The study for all the drug was conducted in Sub Saharan Africa and the number of studies for each drug may be presented in a table form with more cohesive manner as provided in line no 176-178 is very confusing. We think the information is not enough to be placed on a table and we also think there will be too many tables in the manuscript. Many reviews have used our approach in the description of studies. We request to maintain our original flow/style.

2. Author should PCR corrected vs uncorrected provide the others factor responsible for high difference in efficacy among case The reason(s) have been provided in brief as shown in the manuscript

3. The above finding clearly indicated that there is need to monitor the molecular markers of both the drugs and the policy makers may consider this as on priority. Reviewer’s comments have been incorporated in the manuscript 

4. 

5. I believe that parasite clearance time should be tabled as it is important findings.

 Very few studies reported parasite clearance time that’s why we did not include it in our review.

6. I feel that the concluding message from author based on the review and meta-analysis is need more attention for the policy makers as well as the other stakeholders as author fail to discuss the importance of the results of the study. The conclusion has been revised to take on board reviewer’s concern as shown in the manuscript 

 Reviewer 3 

1. The numbers of “Weighted” is hard to read for Fig. 5 Forest plot for dihydroartemisinin -piperaquine PCR unadjusted and adjusted cure rate based. Please, adjust or revise. We have increased the font size for the stated figures. We also have modified other figures for ALU and ASAQ for the sake of uniformity. 

 Editor’s comments 

 1 Please ensure that your manuscript meets PLOS ONE's style requirements, including those for file naming. The PLOS ONE style templates can be found at Editing has been done to meet PLOS ONE style requirements as shown in the manuscript 

2 As noted previously by Academic Editor, your article is similar to the following publication:

https://malariajournal.biomedcentral.com/articles/10.1186/s12936-020-03446-8Please cite and discuss the above study in the introduction and discussion sections of your manuscript, clarifying how the present work is related to the previously published paper.

 The article has been cited as suggested and clarification on how our work is related to the previously published paper has been made

3 Please note that in order to use the direct billing option the corresponding author must be affiliated with the chosen institute. Please either amend your manuscript to change the affiliation or corresponding author, or email us at plosone@plos.org with a request to remove this option. The corresponding author is affiliated with the Catholic University of Health and Allied Sciences as shown in the front page.

4 We note that you have stated that you will provide repository information for your data at acceptance. Should your manuscript be accepted for publication, we will hold it until you provide the relevant accession numbers or DOIs necessary to access your data. Please update our data availability statement. We do not intend to provide accession numbers/DOIs 

5 Please upload a copy of Figure 8, to which you refer in your text on page 15. If the figure is no longer to be included as part of the submission please remove all reference to it within the text. The word figure 8 has been removed as suggested. 

All authors have read, approved the revised manuscript to be submitted to your highly reputable journal. 

It is our hope the manuscript will be suitable for publication. Please contact me if any clarification is still needed.

Sincerely yours,

Karol J. Marwa

Corresponding author.

---

## [Editor Report · Decision Letter 1]

9 Feb 2022

Therapeutic efficacy of Artemether-Lumefantrine, Artesunate-Amodiaquine and Dihydroartemisinin -Piperaquine in the treatment of uncomplicated Plasmodium falciparum malaria in Sub-Saharan Africa: A   systematic review and meta-analysis

PONE-D-21-19495R1

Dear Dr. Marwa,

We’re pleased to inform you that your manuscript has been judged scientifically suitable for publication and will be formally accepted for publication once it meets all outstanding technical requirements.

Kind regards,

Lucy C. Okell

Academic Editor

PLOS ONE
---

## [Editor Report · Acceptance letter]

1 Mar 2022

PONE-D-21-19495R1 

Therapeutic efficacy of Artemether-Lumefantrine, Artesunate-Amodiaquine and Dihydroartemisinin -Piperaquine in the treatment of uncomplicated Plasmodium falciparum malaria in Sub-Saharan Africa: A   systematic review and meta-analysis 

Dear Dr. Marwa:

I'm pleased to inform you that your manuscript has been deemed suitable for publication in PLOS ONE. Congratulations! Your manuscript is now with our production department. 

Kind regards, 

on behalf of

Dr. Lucy C. Okell 

Academic Editor

PLOS ONE